# The Sec61 translocon limits IRE1α signaling during the unfolded protein response

Arunkumar Sundaram[1,2†], Rachel Plumb[1†], Suhila Appathurai[1], Malaiyalam Mariappan[1]*

[1]Department of Cell Biology, Nanobiology Institute, Yale School of Medicine, West Haven, United States; [2]Advance Molecular Biology Lab, School of Health Sciences, University of Science Malaysia, Kubang Kerian, Malaysia

**Abstract** IRE1α is an endoplasmic reticulum (ER) localized endonuclease activated by misfolded proteins in the ER. Previously, we demonstrated that IRE1α forms a complex with the Sec61 translocon, to which its substrate XBP1u mRNA is recruited for cleavage during ER stress (*Plumb et al., 2015*). Here, we probe IRE1α complexes in cells with blue native PAGE immunoblotting. We find that IRE1α forms a hetero-oligomeric complex with the Sec61 translocon that is activated upon ER stress with little change in the complex. In addition, IRE1α oligomerization, activation, and inactivation during ER stress are regulated by Sec61. Loss of the IRE1α-Sec61 translocon interaction as well as severe ER stress conditions causes IRE1α to form higher-order oligomers that exhibit continuous activation and extended cleavage of XBP1u mRNA. Thus, we propose that the Sec61-IRE1α complex defines the extent of IRE1α activity and may determine cell fate decisions during ER stress conditions.

**\*For correspondence:**
malaiyalam.mariappan@yale.edu

[†]These authors contributed equally to this work

**Competing interests:** The authors declare that no competing interests exist.

## Introduction

The majority of secretory and membrane proteins enter the endoplasmic reticulum (ER) through the Sec61 protein translocation channel (*Rapoport, 2007*). In the ER, folding enzymes and chaperones facilitate maturation of newly synthesized proteins. Proteins that fail to achieve their folded state are eliminated by ER-associated quality control pathways (ERAD) (*Brodsky, 2012*), while correctly folded proteins are transported to their intra or extracellular site of activity. When the influx of proteins exceeds the ER protein folding and quality control capacity, misfolded proteins accumulate in the ER leading to a condition known as ER stress. During ER stress, signaling pathways, collectively termed the Unfolded Protein Response (UPR), are activated in order to upregulate chaperones and folding enzymes, to reduce the influx of proteins into the ER, and to increase the capacity for ER-associated degradation (*Walter and Ron, 2011*). In this way, the UPR adapts cells to ER stress conditions and restores ER homeostasis. However, the UPR can also trigger apoptosis during chronic or severe ER stress conditions, suggesting that UPR activity is tightly controlled in order to elicit the appropriate cellular response, whether pro-adaptive or pro-apoptotic (*Hetz, 2012*). Indeed, inappropriate activation of UPR signaling is linked to a number of disease states, including pancreatic beta cell death in diabetes (*Back and Kaufman, 2012*) and neuronal cell death in certain neurodegenerative diseases (*Wang and Kaufman, 2016*).

Three transmembrane sensors, IRE1α, PERK, and ATF6, mediate the UPR. Upon ER stress, all three sensors become activated by changes in their oligomerization state. The most ancient UPR sensor is IRE1α, a transmembrane endonuclease/kinase that senses the accumulation of misfolded proteins in the ER lumen (*Cox et al., 1993*; *Mori et al., 1993*). When ER misfolded proteins are

detected, IRE1α self-oligomerizes through its luminal domains. This, in turn, leads to cytosolic trans-autophosphorylation of the IRE1α kinase domain and subsequent activation of its RNase domain. The activated IRE1α restores the ER folding capacity by cleaving XBP1u mRNA (u; unspliced) to initiate splicing on the ER membrane (*Yoshida et al., 2001*; *Calfon et al., 2002*). Efficient cleavage of XBP1u mRNA requires an interaction between IRE1α and the Sec61 translocon as well as the SRP pathway-mediated recruitment of XBP1u mRNA to the Sec61 translocon (*Plumb et al., 2015*; *Kanda et al., 2016*). Subsequently, the cleaved fragments of XBP1 mRNA are ligated by the RtcB tRNA ligase (*Lu et al., 2014*; *Jurkin et al., 2014*; *Kosmaczewski et al., 2014*) with its co-factor archease (*Poothong et al., 2017*). The spliced XBP1 mRNA is translated into an active transcription factor, XBP1s, which induces UPR genes to alleviate ER stress (*Lee et al., 2003*; *Acosta-Alvear et al., 2007*). In addition, IRE1α also promiscuously cleaves ER-localized mRNAs including mRNAs encoding secretory and membrane proteins, a process known as IRE1α-dependent mRNA decay (RIDD) (*Hollien and Weissman, 2006*; *Hollien et al., 2009*; *Han et al., 2009*). RIDD is implicated in reducing the incoming protein burden on the ER during stress conditions as well as in mediating cell death (*Hollien and Weissman, 2006*; *Ghosh et al., 2014*; *Tam et al., 2014*).

Since the continuous activation of IRE1α is associated with cell death, the activation and inactivation of IRE1α must be properly regulated. Indeed, seminal studies from Peter Walter's group demonstrated that IRE1α activity is temporally and quantitatively attenuated during ER stress conditions (*Lin et al., 2007*). However, the mechanism by which IRE1α is inactivated in the presence of ER stress is unclear. Previous studies have provided important insights into how IRE1α activity can be regulated by its associated proteins (*Bertolotti et al., 2000*; *Okamura et al., 2000*; *Lisbona et al., 2009*; *Eletto et al., 2014*; *Carrara et al., 2015*; *Morita et al., 2017*). Interestingly, factors such as BiP and PDIA6 that are implicated in attenuating IRE1α activity also interact with PERK, which, in contrast to IRE1α, remains activated during prolonged ER stress conditions (*Lin et al., 2007*). We therefore tested the role of the Sec61 translocon in regulating IRE1α activity because it selectively interacts with IRE1α but not with the other ER stress sensors PERK, ATF6 and Ire1β (*Plumb et al., 2015*). We have used a Blue Native polyacrylamide gel electrophoresis (BN-PAGE) immunoblotting procedure to probe IRE1α complexes in cells during normal and ER stress conditions. Our studies reveal that IRE1α exists as preassembled hetero-oligomeric complexes with the Sec61 translocon and becomes activated during ER stress conditions with minor changes to its complexes. We find that the Sec61 translocon limits IRE1α oligomerization and thereby controls activation and inactivation of IRE1α activity during ER stress conditions. Indeed, either the loss of the IRE1α interaction with the Sec61 translocon or severe stress causes IRE1α to form higher-order oligomers that exhibit continuous activation of IRE1α and extended cleavage of XBP1u mRNA. Thus, our studies suggest that the IRE1α-Sec61 complex plays a critical role in controlling IRE1α signaling during ER stress.

## Results

### IRE1α forms hetero-oligomeric complexes with the Sec61 translocon

We hypothesized that the Sec61 translocon may limit IRE1α oligomerization during ER stress and thus control IRE1α activity because of the following observations. First, our previous studies showed that nearly all the endogenous IRE1α is bound with the Sec61 translocon in the ER membrane during normal and ER stress conditions (*Plumb et al., 2015*). Second, the concentration of the Sec61 translocon vastly outnumbers the concentration of IRE1α in the ER (*Plumb et al., 2015*; *Kulak et al., 2014*), suggesting that it could provide a barrier to IRE1α oligomerization. To test this hypothesis, we searched for IRE1α mutants that either disrupt or increase the interaction with the Sec61 translocon. Our previous studies identified a ten amino acid region in the luminal domain proximal to the transmembrane domain of IRE1α that when deleted nearly abolished the interaction with the Sec61 translocon (*Figure 1—figure supplement 1A,B*). We refer to the IRE1α Δ434–443 mutant as weakly interacting IRE1α or wIRE1α. Fortuitously, our previous mutagenesis studies also revealed that IRE1α S439A showed an increased binding to the Sec61 translocon. We then further significantly improved the interaction between IRE1α and Sec61 by combining S439A with the mutation of three hydrophilic residues in the transmembrane domain of IRE1α (*Figure 1—figure supplement 1A,B*) (Sun et al., 2015). We refer to this mutant (IRE1α S439A/T446A/S450A/T451A) as strongly interacting IRE1α (sIRE1α).

To investigate the role of the Sec61 translocon in regulating IRE1α oligomerization and activity, we complemented IRE1α, wIRE1α or sIRE1α into IRE1α-/- HEK 293 Flip-In T-Rex cells generated by CRISPR/Cas9 (*Mali et al., 2013*; *Plumb et al., 2015*). IRE1α expression is controlled by the tetracycline promoter in these complemented cells. Low expression levels as well as ER stress dependent activation of IRE1α were achieved through leaky expression in the absence of doxycycline (*Figure 1A*; *Figure 1—figure supplement 1C*). To examine the oligomerization status of IRE1α in these different cells, we employed a BN-PAGE based immunoblotting procedure. This technique allows separation of large membrane protein complexes with minimal perturbation of native complexes using Coomassie G250 dye as the charged ion carrier (*Wittig et al., 2006*). The cells were treated with or without thapsigargin, which induces ER stress by inhibiting calcium import into the ER lumen, and analyzed by BN-PAGE immunoblotting. Surprisingly, BN-PAGE analysis of IRE1α complemented cells showed two forms of preassembled IRE1α complexes. Form A corresponds to a ~500 kDa complex, and Form B corresponds to a ~720 kDa complex (*Figure 1A*). Intriguingly, upon ER stress treatment, IRE1α Form B slightly increased in intensity, while IRE1α Form A was reduced. Probing phosphorylated IRE1α using the phos-tag reagent (*Yang et al., 2010*) further confirmed that IRE1α was activated upon ER stress as shown by stress dependent detection of phosphorylated IRE1α (*Figure 1A*). To next determine the role of the Sec61 translocon in controlling IRE1α oligomerization, we performed BN-PAGE analysis with cells expressing either wIRE1α, which cannot interact with Sec61, or sIRE1α, which interacts strongly with Sec61 (*Figure 1—figure supplement 1B*). In comparison to the wild-type IRE1α, wIRE1α predominantly existed in the Form B complex, whereas sIRE1α showed significantly more of the Form A (*Figure 1A*). Unlike the wild type, the stress-dependent changes were less obvious for both wIRE1α and sIRE1α oligomers, but they were clearly activated as shown by their phosphorylation using phos-tag based immunoblotting (*Figure 1A*).

Since we did not observe a significant change in IRE1α complexes upon ER stress, we asked if this result was due to a limitation of BN-PAGE to detect changes in IRE1α complexes. To examine this, we performed a BN-PAGE analysis of PERK, the luminal domain of which is structurally similar, and even interchangeable with IRE1α (*Liu et al., 2000*), but does not interact with Sec61 (*Plumb et al., 2015*). Similar to IRE1α, PERK existed as a preformed complex, though of ~900 kDa, in cells under normal conditions. However, upon stress, PERK became a ~1200 kDa complex (*Figure 1B*). These results were recapitulated in HEK293 and insulin secreting rat pancreatic beta-cells (INS-1) treated with ER stress. Here, the endogenous IRE1α again presented as approximately 500 and 720 kDa complexes that changed little during ER stress conditions, while PERK exhibited a significant ER stress-dependent shift in complex size (*Figure 1—figure supplement 2*).

We hypothesized that if the Sec61 translocon controls oligomerization of IRE1α, its depletion in cells should resemble wIRE1α, which exhibited predominantly ~720 kDa complexes on BN-PAGE. Such a result would suggest that the mutation in wIRE1α does not cause secondary effects in IRE1α independent of the Sec61 translocon interaction disruption. To test this, we depleted the Sec61 translocon by treating cells with siRNA oligos against Sec61α and performed BN-PAGE analysis. Remarkably, wild-type IRE1α resembled wIRE1α in the Sec61 translocon depleted cells, as the 500 kDa Form A shifted to the 720 kDa Form B (*Figure 1C*). In contrast, the Sec61 translocon depletion had little effect in cells expressing wIRE1α, which remained in Form B. Consistent with recent findings (*Adamson et al., 2016*), depletion of the Sec61 translocon partially activated IRE1α as shown by a slight increase in self-phosphorylation in the absence of ER stress compared to control siRNA treated cells. However, an efficient activation of IRE1α in these cells typically required treatment with the ER stress inducer thapsigargin (*Figure 1C*). Intriguingly, the depletion of the Sec61 translocon specifically affected IRE1α complexes, as PERK complexes were less disrupted in Sec61 depleted cells relative to the control siRNA-depleted cells (*Figure 1D*). To determine if the Sec61 translocon co-migrates with IRE1α complexes, we performed BN-PAGE immunoblotting with Sec61α antibodies. The Sec61 translocon, which is composed of α, β, and γ subunits, ran predominantly as a ~146 kDa form and a minor ~350 kDa form on BN-PAGE (*Figure 1E*, *Figure 1—figure supplement 3*), which is consistent with previous studies (*Conti et al., 2015*). Currently, it is unclear why we were not able to detect Sec61 co-migration with IRE1α, though it is likely that only a small population of the highly abundant Sec61 exists in a complex with IRE1α in cells. Collectively, these results suggest that IRE1α complexes in cells are regulated by an interaction with the Sec61 translocon.

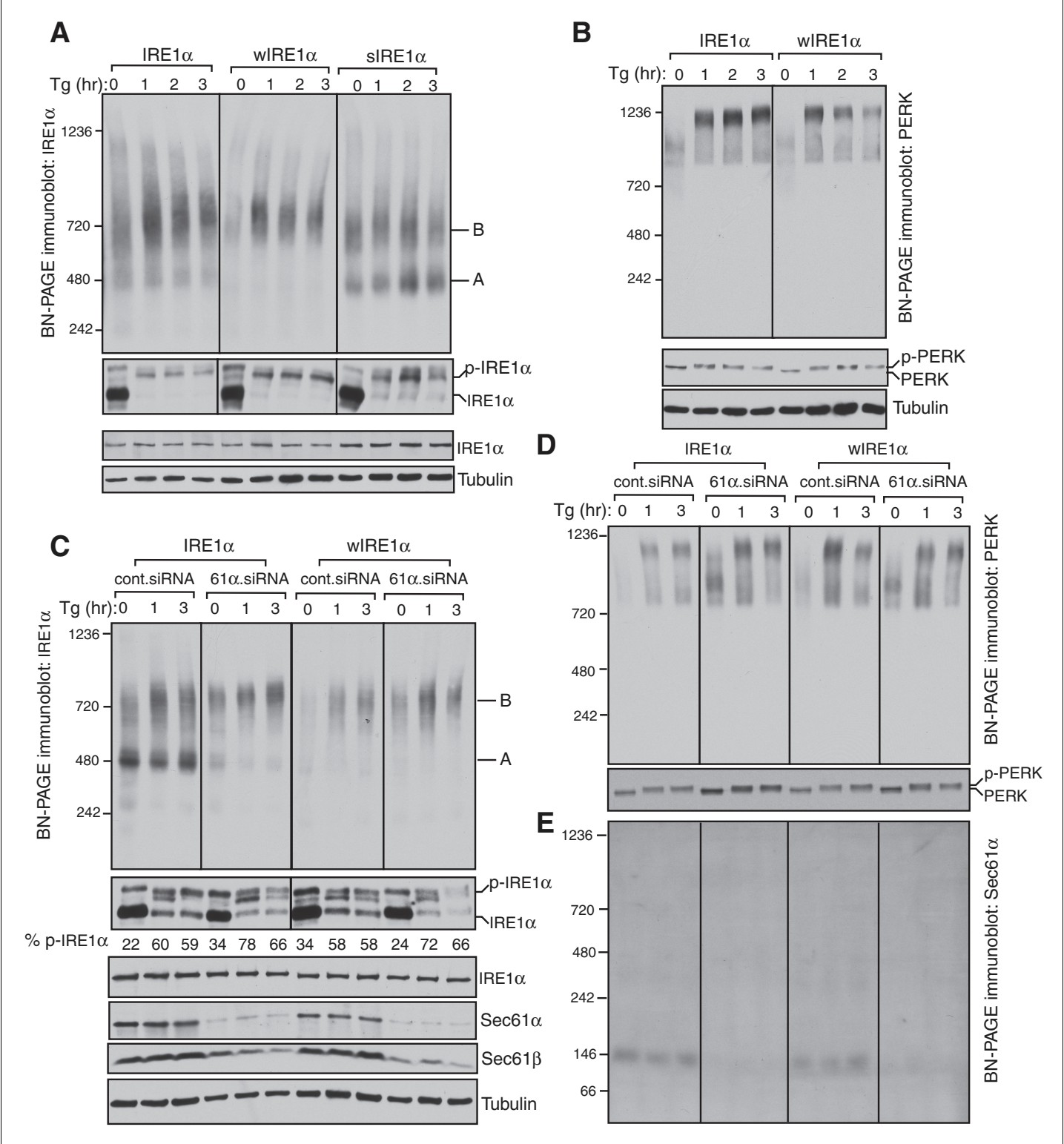

**Figure 1.** IRE1α complexes are regulated by an interaction with the Sec61 translocon. (**A**) IRE1α -/- HEK293 cells complemented with wild-type IRE1α-HA, wIRE1α-HA (Δ434–443), or sIRE1α-HA (S439A/T446A/S450A/T451A) were treated with 2.5 µg/ml thapsigargin (Tg) for the indicated hours (hr), lysed with digitonin, and analyzed by BN-PAGE immunoblotting (top) as well as phos-tag based immunoblotting to probe phosphorylated IRE1α (bottom). *A* denotes a ~500 kDa complex of IRE1α in BN-PAGE immunoblotting. *B* denotes a ~720 kDa complex of IRE1α. (**B**) The cells expressing IRE1α-HA or wIRE1α-HA were treated with 2.5 ug/ml Tg for the indicated hours and analyzed by both BN-PAGE immunoblotting and standard immunoblotting with a PERK antibody. (**C**) IRE1α-HA or wIRE1α-HA expressing cells were treated with either control siRNA or Sec61α siRNA followed by treatment with 2.5

*Figure 1 continued on next page*

*Figure 1 continued*

μg/ml Tg for the indicated times. The samples were analyzed as in panel **A**. (**D,E**) The samples from the panel **C** were analyzed by BN-PAGE immunoblotting with either PERK or Sec61α antibodies.

The following figure supplements are available for figure 1:

**Figure supplement 1.** IRE1α mutants that either disrupt the interaction or improve the interaction with Sec61 translocon.

**Figure supplement 2.** Endogenous IRE1α exists as preformed complexes in HEK293 and INS-1 cells.

**Figure supplement 3.** BN-PAGE analysis of the Sec61 translocon.

We next sought to determine whether the Sec61 translocon co-migrates with the different complexes of IRE1α by performing BN-PAGE with the purified Sec61-IRE1α complex. To achieve this, we established stable cell lines expressing IRE1α and purified the IRE1α and Sec61 complex through a combination of affinity and ion exchange chromatography using digitonin, which preserves the interaction between IRE1α and the Sec61 translocon. The coomassie blue stained gel revealed that purified IRE1α associated with the Sec61 translocon and Sec63, a component of the translocon complex (*Figure 2A*) (*Meyer et al., 2000*). As expected, wIRE1α lacked the Sec61 translocon complex, whereas sIRE1α associated with the Sec61 translocon complex (*Figure 2A*). All three IRE1α proteins had a similar ability to cleave in vitro transcribed XBP1u mRNA substrate, though wIRE1α and sIRE1α showed slightly slower kinetics of cleavage (*Figure 2—figure supplement 1*).

We then analyzed these purified proteins by BN-PAGE immunoblotting to determine if the Sec61 translocon co-migrates with different IRE1α complexes. Similar to the IRE1α complexes in cells, purified IRE1α existed as complexes of both Form A and Form B when it associated with the Sec61 translocon (*Figure 2B*). In contrast, the purified wIRE1α existed predominantly as Form B and as a ~240 kDa complex. The 240 kDa form of IRE1α was not obvious in cells, suggesting that IRE1α complexes may be labile during the purification procedure. sIRE1α closely resembled the wild-type IRE1α complexes because both purified IRE1α and sIRE1α proteins contained similarly enriched Sec61 translocon complex (*Figure 2C*). Remarkably, BN-PAGE analysis with Sec61α antibodies revealed that Sec61 co-migrates with both Form A and Form B in purified IRE1α and sIRE1α (*Figure 2C*). In contrast, Sec61α was not detectable in BN-PAGE with the purified wIRE1α. At present, the role of BiP, which is known to interact and inhibit IRE1α oligomerization (*Bertolotti et al., 2000*; *Okamura et al., 2000*; *Oikawa et al., 2009*; *Carrara et al., 2015*), in the Sec61 translocon-mediated regulation of IRE1α complexes is unclear, since we could not detect BiP in our purified IRE1α complexes (*Figure 2D*). Nevertheless, our results with purified IRE1α proteins are consistent with the results derived from cells. We find that IRE1α and sIRE1α exist in Forms A and B with the Sec61 translocon, while wIRE1α is predominantly in Form B but without the Sec61 translocon. Although further work is required to determine the precise copy numbers of IRE1α in these complexes, our data suggest that Sec61 is an intrinsic part of the IRE1α complexes under normal and ER stress conditions.

## The Sec61 translocon inhibits formation of higher order IRE1α oligomeric clusters in cells

We next asked whether the Sec61 translocon-mediated regulation of IRE1α oligomerization can be observed by immunofluorescence. Previous studies reported that IRE1α forms higher-order oligomers or clusters upon ER stress, which correlate with IRE1α RNase activity and are proposed to be important for IRE1α signaling (*Li et al., 2010*). To determine whether the Sec61 translocon mediates regulation of IRE1α oligomerization, we looked for ER stress-dependent changes in IRE1α oligomerization in IRE1α-/- HEK293 cells complemented with IRE1α variants containing a C-terminal HA tag to facilitate immunostaining. Under normal conditions, IRE1α and wIRE1α were diffusely distributed in the ER membrane and colocalized with Sec61β, a subunit of the Sec61 translocon (*Figure 3—figure supplement 1*). Strikingly, we detected robust clusters with wIRE1α expressing cells but not in wild-type IRE1α expressing cells upon treatment with tunicamycin, which induces ER stress by inhibiting protein glycosylation in the ER (*Figure 3A*; *Figure 3—figure supplement 1*). Similar to wild-

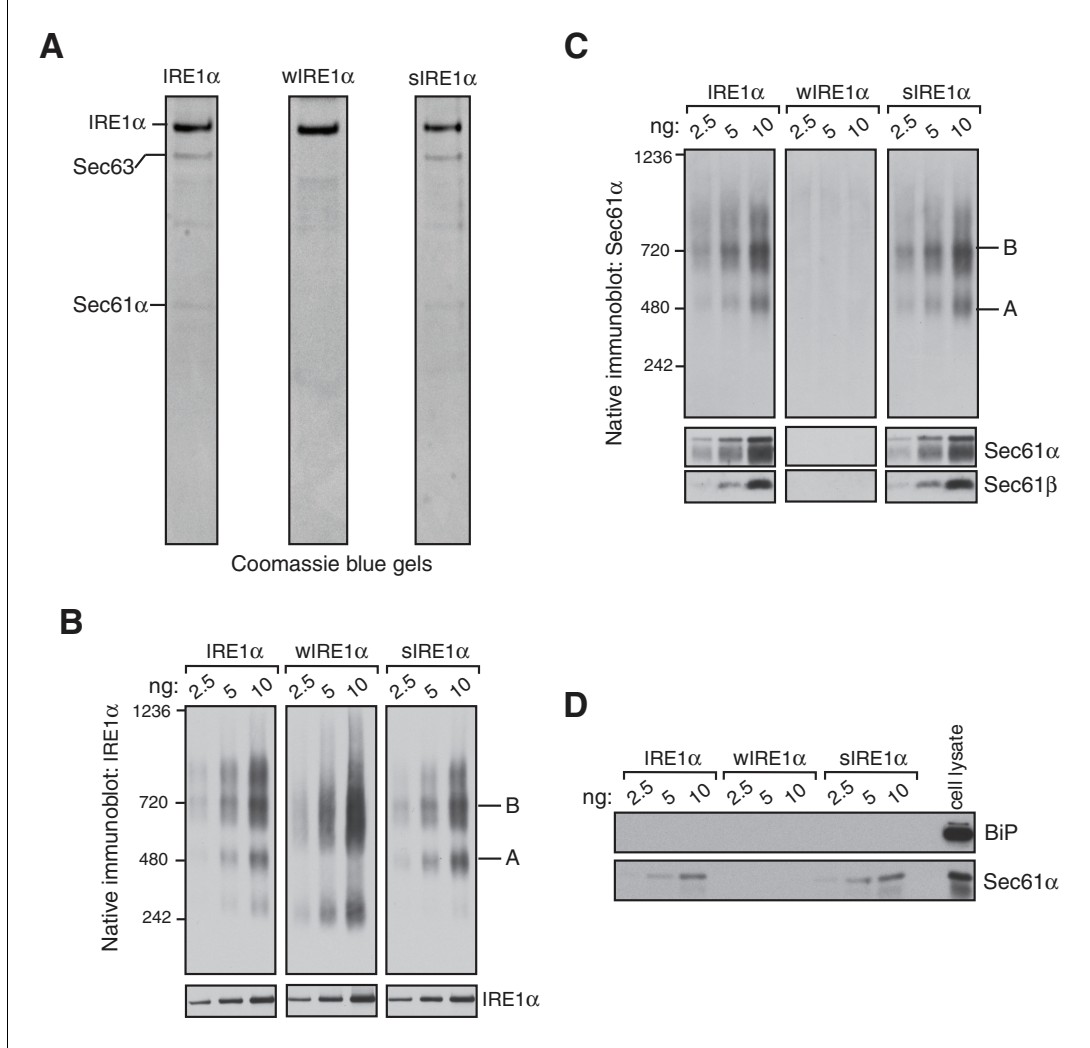

**Figure 2.** IRE1α forms a hetero-oligomeric complex with the Sec61 translocon. (**A**) Coomassie blue stained gels showing IRE1α variants that were purified from HEK293 cells stably expressing 2X strep-tagged IRE1α. (**B**) The indicated concentration of purified IRE1α proteins was analyzed by BN-PAGE based immunoblotting with IRE1α antibodies. (**C**) The purified IRE1α proteins were analyzed as in panel **B** using Sec61α antibodies. (**D**) The purified IRE1α proteins were analyzed by standard immunoblotting with BiP and Sec61α antibodies.

The following figure supplement is available for figure 2:

**Figure supplement 1.** XBP1u mRNA cleavage by purified IRE1α variants.

type IRE1α, we failed to observe clusters in sIRE1α expressing cells (**Figure 3A**), supporting the idea that the IRE1α interaction with the Sec61 translocon limits cluster formation. wIRE1α clustering was not dependent on cell types since we obtained similar results when we analyzed IRE1α-/- mouse embryonic fibroblast (MEF) cells complemented with either wild-type or wIRE1α (**Figure 3—figure supplement 2A**). In addition, ER stress-mediated clusters of wIRE1α were not unique to this particular wIRE1α mutant, which has a ten amino acid deletion in IRE1α, but were also observed in cells expressing a wIRE1α mutant where two critical residues are mutated within the ten amino acid region (**Figure 3—figure supplement 2B**). Only after increasing the expression level of IRE1α using doxycycline, could we detect a small percentage of clusters in wild-type IRE1α expressing cells (**Figure 3B,C,D**). In contrast, we detected robust wIRE1α clusters in ER stress treated cells even at low expression levels. Together, these results suggest that the Sec61 translocon interaction prevents

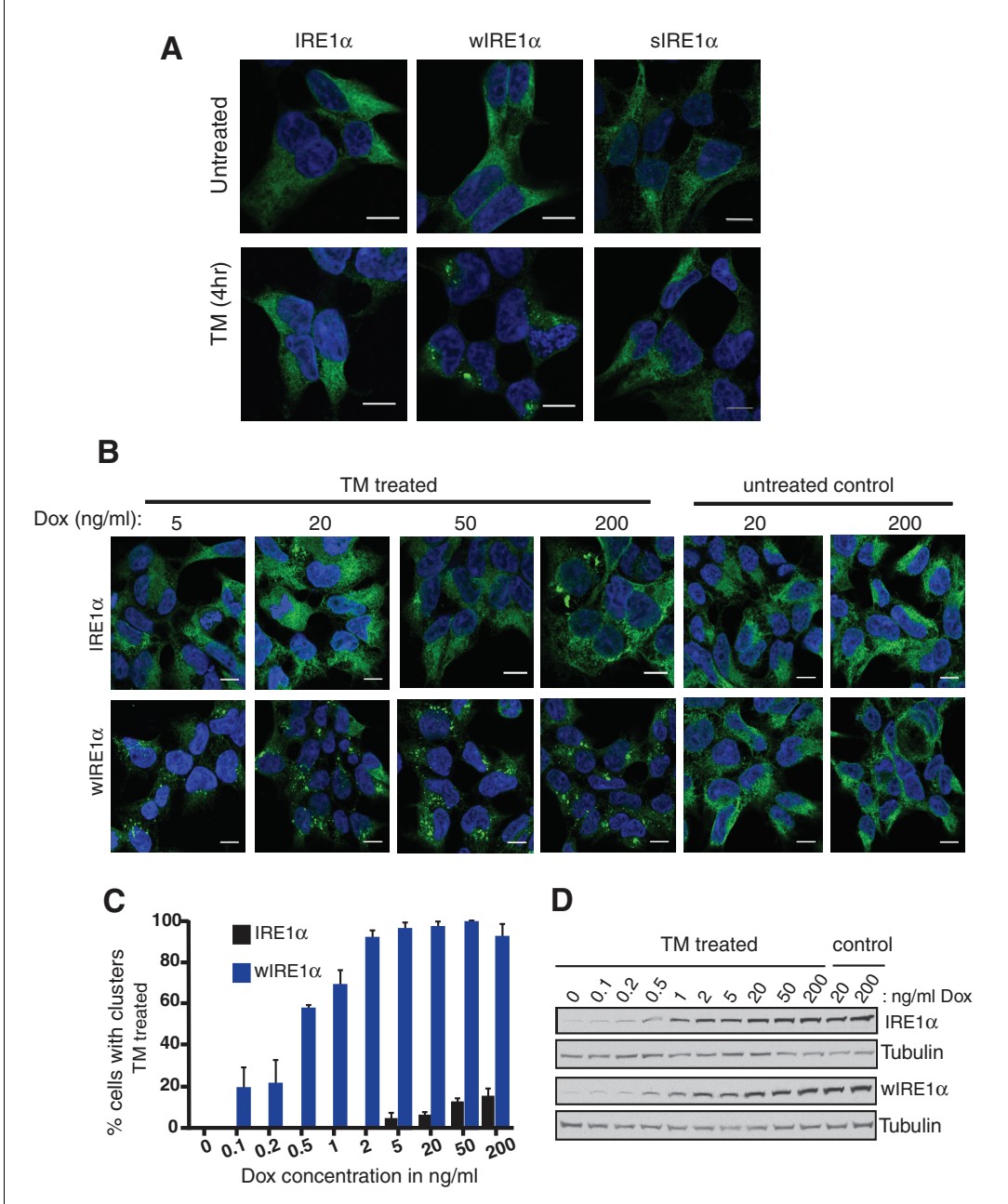

**Figure 3.** The Sec61 translocon inhibits IRE1α higher-order oligomer or cluster formation in cells. (**A**) IRE1α -/- HEK293 cells complemented with IRE1α-HA, wIRE1α-HA or sIRE1α-HA were treated with 5 µg/ml Tunicamycin (TM) for 4 hr. Scale bars are 10 µm. Subsequently, cells were processed using an immunostaining procedure to label IRE1α (green) with rabbit anti-HA antibodies as well as a Hoechst stain to label nuclei (blue) and imaged using a confocal microscope. (**B**) IRE1α-HA or wIRE1α-HA expressing cells were induced with various amounts of doxycycline, treated with TM and analyzed as in panel A. (**C**) Quantification of the number of cells with IRE1α clusters from the panel C. Error bar represents standard deviation. (**D**) Immunoblots show the expression of IRE1α in response to varying concentrations of doxycycline.

The following source data and figure supplements are available for figure 3:

**Source data 1.** Doxycycline titration and quantification of IRE1α clusters as described *Figure 3C*.

**Figure supplement 1.** IRE1α and wIRE1α are localized to the ER in HEK293 cells.

**Figure supplement 2.** The Sec61 translocon interaction defective IRE1α mutant form clusters in both MEF and HEK293 cells.

the formation of IRE1α higher order oligomers or clusters in cells. In contrast with previous work (*Li et al., 2010*; *Ghosh et al., 2014*), we observe only a low percentage of cells containing wild-type IRE1α clusters. This difference may be due to the intensity of ER stress applied to monitor IRE1α clusters in cells. Nevertheless, the differences we observe between wild-type IRE1α and wIRE1α indicate that the Sec61 translocon inhibits the formation of these higher-order oligomers or clusters.

## Proper activation of IRE1α relies on the interaction between IRE1α and the Sec61 translocon

Since wIRE1α robustly formed higher order oligomeric clusters under ER stress conditions, we predicted that it may be more quickly activated than wild-type IRE1α. To test this idea, we gradually increased the expression level of IRE1α by titrating the concentration of doxycycline and assayed for activation by probing for IRE1α phosphorylation (*Figure 4A,B*). Consistent with previous findings, overexpressed IRE1α was partially activated as shown by phosphorylation even in the absence of ER stress (*Li et al., 2010*). Overexpressed wIRE1α exhibited an even larger amount of auto-phosphorylation and thus activation compared to wild-type IRE1α, while overexpressed sIRE1α showed

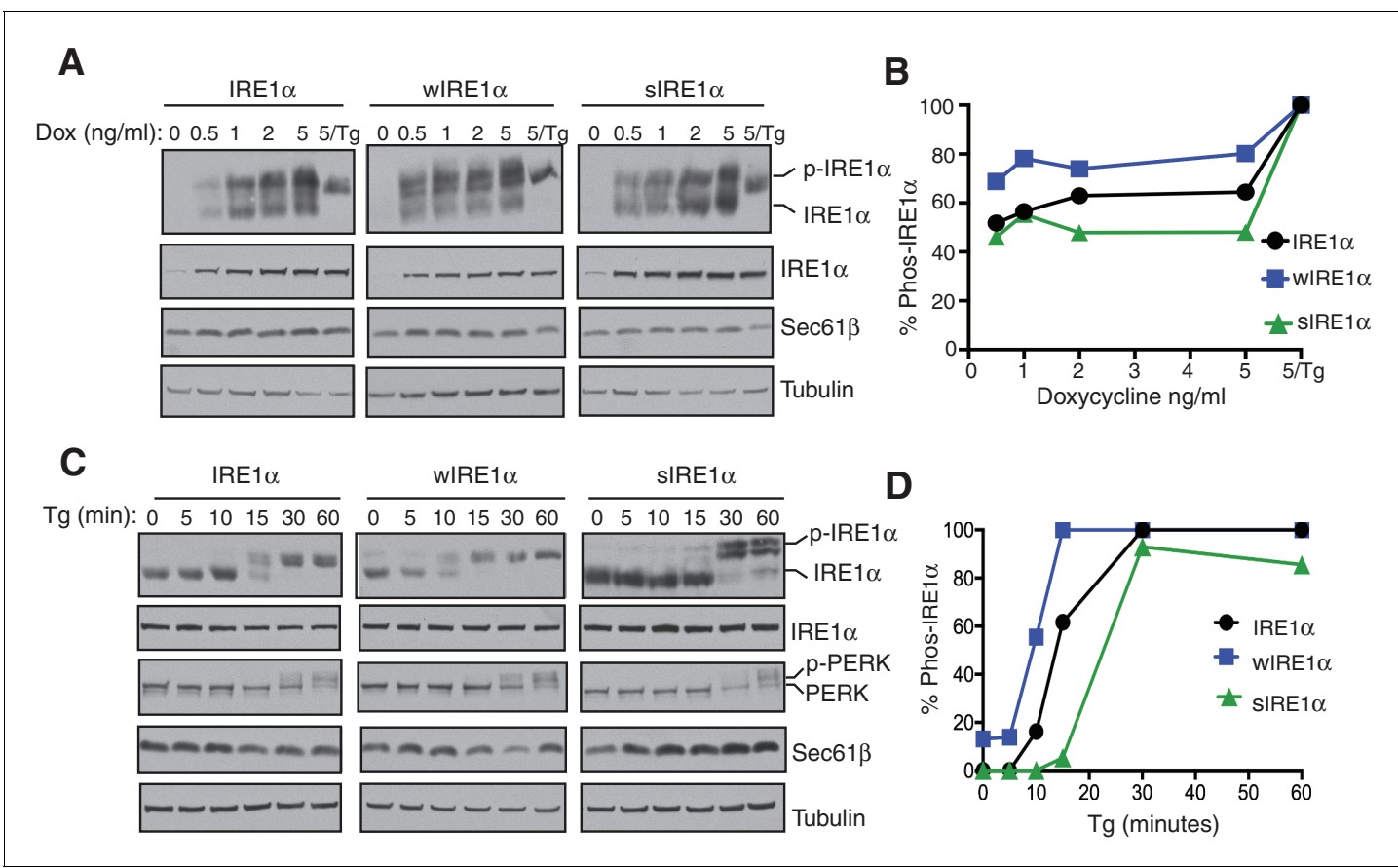

**Figure 4.** The Sec61 translocon regulates the activation of IRE1α during ER stress. (**A**) IRE1α -/- HEK293 cells complemented with either wild type IRE1α -HA, wIRE1α-HA, or sIRE1α-HA were induced with the indicated amounts of doxycycline, treated with 2.5 μg/ml Tg for 2 hr where indicated and analyzed by phos-tag immunoblotting for IRE1α and standard immunoblotting for the indicated antigens. (**B**) Quantification of IRE1α, wIRE1α, and sIRE1α phosphorylation from panel **A**. (**C**) IRE1α-HA, wIRE1α-HA, or sIRE1α-HA expressing cells were treated with 1 μg/ml of Tg for the indicated time points and analyzed as in panel **A**. (**D**) Quantification of IRE1α, wIRE1α, and sIRE1α phosphorylation from panel **C**.

The following source data is available for figure 4:

**Source data 1.** Doxycycline titration and activation of IRE1α, wIRE1α or sIRE1α as described *Figure 4B*.
**Source data 2.** Activation of IRE1α, wIRE1α or sIRE1α in Tg-treated cells as described *Figure 4D*.

reduced auto-phosphorylation compared to wild-type IRE1α. Interestingly, all IRE1α variants required ER stress treatment, in this case thapsigargin, to achieve a full activation state, suggesting that the accumulation of misfolded proteins plays a major role in IRE1α activation (*Figure 4A,B*). We next tested the role of the Sec61 translocon in IRE1α activation during ER stress treatment. Consistently, wIRE1α was more quickly activated as shown by auto-phosphorylation compared to the wild type, whereas sIRE1α was activated at slower rate during ER stress (*Figure 4C,D*). As a control, we probed for the activation of PERK, which was activated similarly in all three IRE1α variants expressing cells. Taken together, our results suggest that the proper activation of IRE1α relies on an interaction with the Sec61 translocon.

## The attenuation of IRE1α signaling requires an interaction with the Sec61 translocon

We reasoned that higher order oligomers and clusters of IRE1α formed by disrupting the IRE1α-Sec61 translocon interaction might be altering the inactivation rate of IRE1α during ER stress. Therefore, we compared ER stress-induced inactivation of IRE1α and wIRE1α by probing for IRE1α phosphorylation. IRE1α was fully activated after two hours of ER stress treatment as demonstrated by all IRE1α shifting to the phosphorylated state (*Figure 5A,B*). Spliced XBP1 (XBP1s) protein production peaked at five hours. During prolonged stress, IRE1α was gradually inactivated with a concomitant reduction in the production of spliced XBP1 protein (XBP1s) (*Figure 5A*). Unlike IRE1α, PERK was activated through the duration of the stress period. This is consistent with previous studies which showed that IRE1α-mediated XBP1u mRNA splicing diminished within a few hours of stress despite the continuation of the ER stress treatment (*Lin et al., 2007*). In sharp contrast to wild-type IRE1α, the Sec61 interaction-defective mutant, wIRE1α, showed significantly reduced inactivation as well as extended production of XBP1s during prolonged ER stress (*Figure 5A,B*). A similar difference in IRE1α and wIRE1α phosphorylation was observed with tunicamycin (*Figure 5C,D*). Here, IRE1α was nearly completely inactivated, but wIRE1α was only partially inactivated during prolonged stress. The temporal inactivation of IRE1α during ER stress was not specific to the complemented recombinant IRE1α since we obtained a similar result with the endogenous IRE1α in HEK293 cells (*Figure 5—figure supplement 1*). Consistent with our previous work, under ER stress treatment conditions when both IRE1α and wIRE1α are equally activated, wIRE1α cells produced less XBP1s protein (*Figure 5A,C and* five hour treatment) since the lack of the Sec61 translocon interaction prevented efficient XBP1u mRNA cleavage (*Plumb et al., 2015*). We therefore wondered whether the slow attenuation observed in wIRE1α expressing cells was due to decreased XBP1s production, which could cause a reduction in ER chaperone production. However, we found that production of XBP1s-induced proteins, such as BiP and the Sec61 translocon, were similar in both IRE1α and wIRE1α expressing cells (*Figure 5A,C*). To further confirm that XBP1s levels were not causing the observed phenotype, we overexpressed XBP1s by transfecting an XBP1s expressing plasmid into both IRE1α and wIRE1α expressing cells. Despite the overexpression of XBP1s, wIRE1α was still attenuated significantly slower than wild-type IRE1α (*Figure 5E,F*).

We predicted that if the Sec61 translocon promotes IRE1α inactivation, sIRE1α, which interacts strongly with Sec61, should be more quickly inactivated than the wild type IRE1α. Indeed, sIRE1α showed a faster inactivation rate during prolonged ER stress conditions (*Figure 5G,H*). Finally, we asked whether the presence of misfolded proteins in the ER is required for the continuous activation of wIRE1α during ER stress. Halting protein synthesis after removing ER stress allowed for complete inactivation of wIRE1α similar to IRE1α and PERK (*Figure 5—figure supplement 2*). This result implies that the presence of misfolded proteins in the ER is required for the continuous activation of wIRE1α. Together, these results indicate that an efficient inactivation of IRE1α requires the IRE1α interaction with the Sec61 translocon.

## Severe ER stress induces clusters and extended activation of wild-type IRE1α

Our results suggested that the Sec61 translocon limits IRE1α oligomerization and thereby controls activation and inactivation of IRE1α during ER stress. Therefore, we hypothesized that severe ER stress may overcome this restriction and induce higher-order oligomers as well as extended activation of IRE1α in wild-type IRE1α expressing cells, similar to that observed with wIRE1α. To test this,

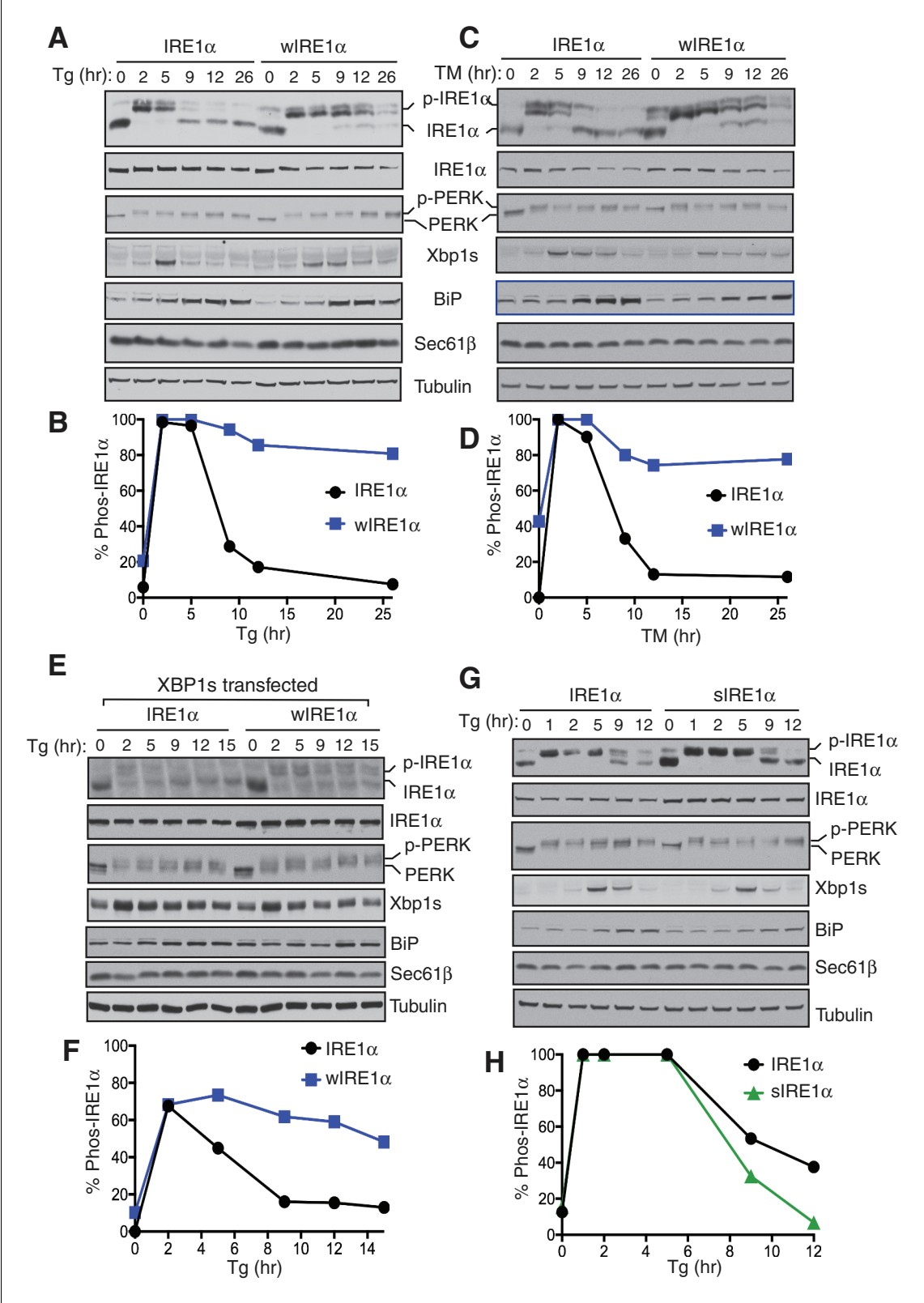

**Figure 5.** The Sec61 translocon regulates the attenuation of IRE1α activity during ER stress. (**A**) IRE1α -/- HEK293 cells complemented with either wild type IRE1α-HA or wIRE1α-HA were treated with 2.5 µg/ml of Tg for the indicated time points and analyzed by phos-tag immunoblotting for IRE1α and standard immunoblotting for the indicated antigens. (**B**) Quantification of IRE1α and wIRE1α phosphorylation from panel **A**. (**C**) IRE1α-HA or wIRE1α-HA cells were treated with 10 µg/ml of TM for the indicated time points and analyzed as in panel **A**. (**D**) Quantification of IRE1α and wIre1 phosphorylation

*Figure 5 continued on next page*

*Figure 5 continued*

from panel **C**. (**E**) IRE1α-HA or wIRE1α-HA cells were transfected with XBP1s plasmid and treated with 1 µg/ml of Tg for the indicated time points and analyzed as in panel **A**. (**F**) Quantification of IRE1α and wIRE1α phosphorylation from panel **E**. (**G**) IRE1α-HA or sIRE1α-HA cells were treated with 2.5 µg/ml of Tg for the indicated time points and analyzed as in panel **A**. (**H**) Quantification of IRE1α and sIRE1α phosphorylation from panel **G**.

The following source data and figure supplements are available for figure 5:

**Source data 1.** Attenuation of IRE1α and wIRE1α in Tg-treated cells as described in *Figure 5B*.

**Source data 2.** Attenuation of IRE1α and wIRE1α in TM-treated cells as described *Figure 5D*.

**Source data 3.** Attenuation of IRE1α and wIRE1α in XBP1s expressing cells as described *Figure 5F*.

**Source data 4.** Attenuation of IRE1α and sIRE1α in Tg-treated cells as described in *Figure 5H*.

**Figure supplement 1.** Attenuation of the endogenous IRE1α activity during ER stress.

**Figure supplement 2.** Accumulation of misfolded proteins is required for the activation of IRE1α.

we examined IRE1α cluster formation after increasing the intensity of ER stress by adding four-fold more thapsigargin. This high concentration of thapsigargin, but not a lower concentration, induced clusters in IRE1α, wIRE1α, and sIRE1α expressing cells, though a higher percentage of wIRE1α cells presented clusters than wild-type IRE1α or sIRE1α cells (*Figure 6A,B*). These results suggest that the Sec61-IRE1α complex plays a role in limiting IRE1α oligomerization under ER stress conditions, but increased misfolded protein accumulation during severe ER stress conditions overcomes the Sec61 translocon-mediated restriction of IRE1α oligomerization. Interestingly, the interaction between IRE1α and the Sec61 translocon was little changed during both medial and severe ER stress conditions (*Figure 6—figure supplement 1*). We therefore hypothesized that the Sec61 translocon might be clustering with IRE1α during severe ER stress conditions. Consistent with our hypothesis, confocal imaging revealed that the endogenous Sec61 translocon co-localized with IRE1α clusters in both wild type IRE1α and sIRE1α expressing cells. However, wIRE1α clusters appear to lack the Sec61 translocon (*Figure 6—figure supplement 2*).

Since wild type IRE1α resembles wIRE1α in forming higher order oligomers or clusters during severe ER stress conditions, we predicted that wild-type IRE1α deactivation might also resemble wIRE1α under such conditions. Indeed, the attenuation of wild-type IRE1α during severe stress was significantly delayed compared to less severe stress. Thus, the production of spliced XBP1 mRNA and its protein were continued (*Figure 6C,D and E*). These results suggest that once IRE1α forms higher oligomers, due to either a defect in the interaction with Sec61 or under severe ER stress, it becomes resistant to inactivation.

## Discussion

In this study, we addressed the question of how IRE1α activity is regulated during ER stress conditions. We find that IRE1α oligomerization and RNAse activity are limited by the Sec61 translocon during normal and remedial ER stress levels, but that severe ER stress overcomes this block, resulting in prolonged IRE1α activation. Our results point to an important role for the IRE1α-Sec61 complex in measuring ER stress levels and accordingly tuning IRE1α activity, which may determine cell fate during ER stress.

To determine the role of the Sec61 translocon in regulating IRE1α oligomerization in cells under normal and ER stress conditions, we employed a BN-PAGE immunoblotting protocol. To our surprise, we found that IRE1α appears to be in preassembled complexes during steady-state conditions. Upon ER stress, the IRE1α complexes showed little change, albeit the intensity of Form B slightly increased with stress. This result suggests that IRE1α activation is most likely caused by a conformational change induced within the preformed IRE1α complexes by binding with misfolded proteins in the lumen. In contrast, changes in the PERK complex were conspicuous upon ER stress,

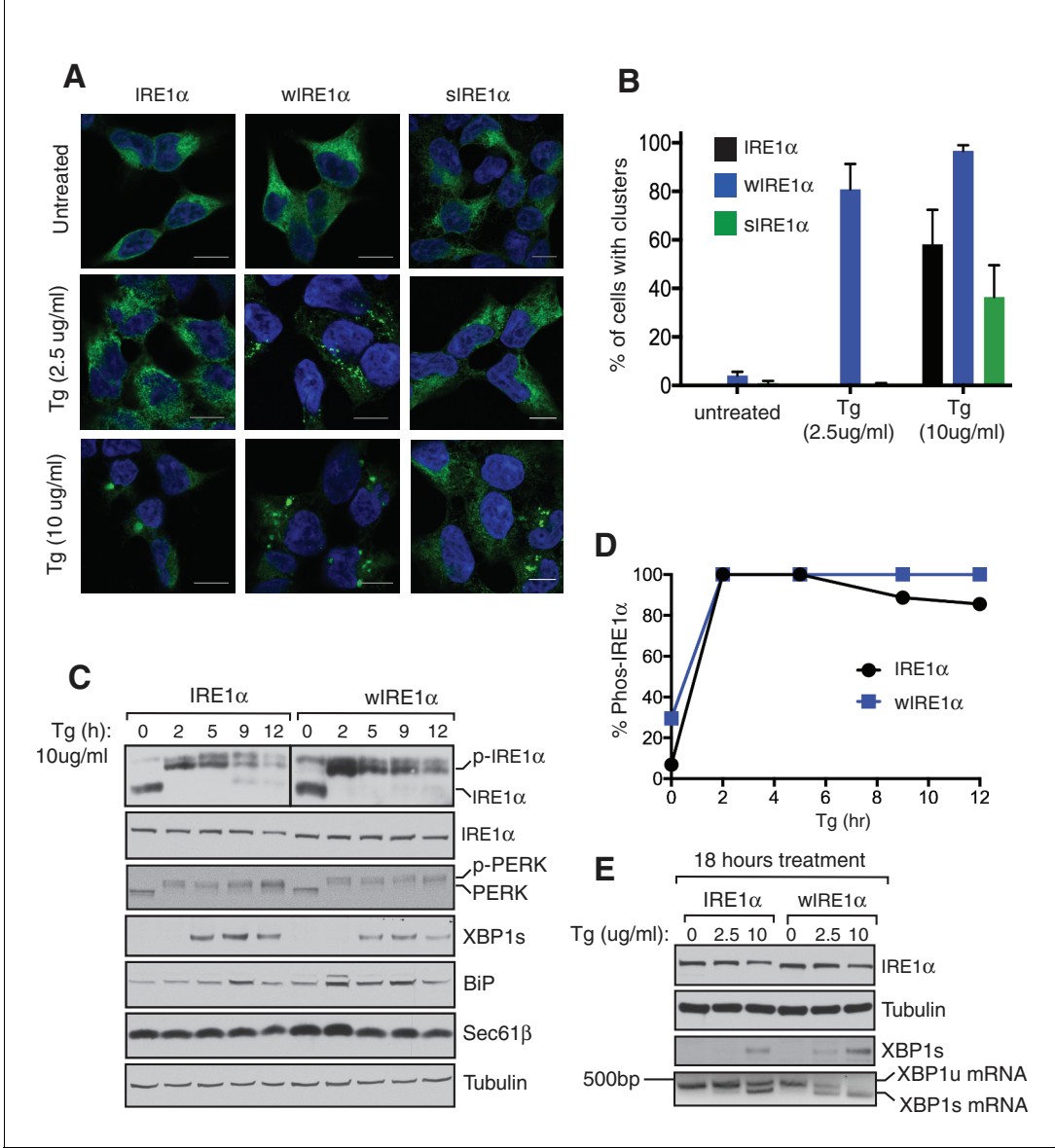

**Figure 6.** Severe ER stress causes higher-order oligomer formation and extended activation of wild type IRE1α. (A) IRE1α-HA, sIRE1α-HA, and wIRE1α-HA complemented IRE1α -/- HEK293 cells were treated with 2.5 µg/ml Tg for 4 hr or 10 µg/ml Tg for 2 hr. Subsequently, cells were processed using an immunostaining procedure to label IRE1α (green) with rabbit anti-HA as well as a Hoechst stain to label nuclei (blue). Scale bars are 10 µm. (B) Images from A were analyzed to determine the number of cells containing IRE1α, wIRE1α clusters, or sIRE1α clusters. Error bar represents S.E.M. (C) IRE1α-HA or wIRE1α-HA expressing cells were treated with 10 µg/ml Tg for the indicated time points and analyzed by phos-tag immunoblotting for IRE1α and standard immunoblotting for the indicated antigens. (D) Quantification of IRE1α and wIRE1α phosphorylation from panel C. (E) IRE1α or wIRE1α expressing cells were treated with either 2.5 µg/ml Tg or 10 µg/ml Tg for 18 hr and analyzed by immunoblots as well as the XBP1 mRNA splicing assay. XBP1u - Unspliced XBP1 mRNA, XBP1s - spliced XBP1 mRNA.

The following source data and figure supplements are available for figure 6:

**Source data 1.** Quantification of IRE1α clusters under sever stress as described *Figure 6B*.
**Source data 2.** Attenuation of IRE1α or wIRE1α under severe stress as described *Figure 6D*.
**Figure supplement 1.** The IRE1α interaction with the Sec61 translocon is stable during severe ER stress conditions.
**Figure supplement 2.** Co-localization of IRE1α and Sec61 during severe ER stress.

as it moves from ~720 kDa to ~1200 kDa in size. These results led us to wonder what advantage preassembled complexes of IRE1α might have over ER stress-induced IRE1α oligomers. We propose that the extreme low-abundance of IRE1α (*Plumb et al., 2015*; *Kulak et al., 2014*) might result in a very slow rate of oligomer formation and activation. Thus, preassembled IRE1α complexes may be essential for the rapid and robust IRE1α activation observed in cells. Future work is required to determine how many IRE1α molecules are present in each complex of IRE1α on BN-PAGE.

We next investigated higher-order oligomerization of IRE1α by examining cluster formation during ER stress. It has been reported that IRE1α forms clusters upon ER stress that correspond to higher-order oligomers (*Li et al., 2010*). In accordance with previous reports, we find that wild-type IRE1α forms clusters, though we only observe significant cluster formation under severe ER stress conditions. In contrast, wIRE1α formed robust clusters during remediable ER stress conditions and exhibited a higher percentage of clusters than IRE1α during severe stress conditions. These data suggest that the Sec61 translocon limits IRE1α cluster formation and that the preassembled complexes of wIRE1α may collide and form clusters rapidly in the absence of the Sec61 interaction. Through this method, we observed large, ER stress dependent, IRE1α clusters that were not captured in our BN-PAGE assay. The precise reason for this is not well understood at this point, although we cannot exclude the limitation of BN-PAGE in detecting the transient and highly dynamic nature of higher-order oligomers or clusters of IRE1α (*Li et al., 2010*).

Severe ER stress drastically increases cluster formation in cells expressing wild-type IRE1α, suggesting that the Sec61 translocon-mediated restriction of IRE1α oligomerization may be overcome under these conditions. Exactly how severe stress precisely tempers the Sec61 translocon barrier warrants further investigation. One potential explanation is that severe ER stress increases the number of misfolded polypeptides in the ER, which may overcome the Sec61 barrier and drive the IRE1α and Sec61 complexes into clusters. This is supported by previous studies that indicate misfolded proteins can directly bind and activate yeast IRE1α (*Kimata et al., 2007*; *Gardner and Walter, 2011*), though the similar evidence is currently lacking in metazoans (*Oikawa et al., 2012*). Future structural and biochemical studies are needed to understand how the Sec61 translocon is precisely arranged with IRE1α to prevent IRE1α oligomerization and how this barrier is overcome under severe ER stress conditions.

Our studies also revealed that IRE1α interaction with Sec61 might be necessary to prevent inappropriate activation during physiological low levels of stress. This is apparent with wIRE1α, where a small population is constitutively activated in the absence of stress, and overall it presents increased ER stress sensitivity and exhibits prolonged activity. The constitutive activation of wIRE1α under basal conditions is consistent with the recent findings that IRE1α signaling is activated upon depletion of the Sec61 translocon in cells (*Adamson et al., 2016*). These findings fit with the intriguing model proposed where IRE1α may sense the Sec61 translocon level and accordingly upregulate Sec61 genes by cleaving XBP1u mRNA (*Adamson et al., 2016*). However, it remains to be understood how the low abundant IRE1α becomes activated by subtle quantity changes in vastly more abundant Sec61 translocon.

At present, the role of BiP in the Sec61 mediated regulation of IRE1α oligomerization is unclear. Similar to wIRE1α, earlier studies have shown that a small fraction of the BiP interaction defective IRE1α mutant is constitutively activated even under normal conditions as reflected by XBP1u mRNA cleavage (*Oikawa et al., 2009*). Therefore, it is likely that both BiP and the Sec61 translocon are required to maintain IRE1α in an inactive form under normal conditions. However, unlike BiP, which is released from IRE1α during ER stress (*Bertolotti et al., 2000*; *Okamura et al., 2000*; *Oikawa et al., 2009*; *Pincus et al., 2010*), the interaction with the Sec61 translocon is maintained throughout ER stress (*Figure 6—figure supplements 1* and *2*). We therefore propose that the Sec61 translocon may play a crucial role during ER stress to limit IRE1α activity. The disparate effects of the wIRE1α and sIRE1α mutants, which either promote or prevent IRE1α oligomerization and activation, respectively, lend support for this model.

In comparison to wild-type IRE1α, the attenuation of wIRE1α activity is significantly delayed. One plausible explanation is that wIRE1α robustly forms large oligomers that sustain a longer activation period during ER stress. In contrast, sIRE1α was slower to phosphorylate and more quickly attenuated than wild-type IRE1α. It is likely that other IRE1α interacting proteins besides Sec61 also contribute to IRE1α activation and attenuation (*Lisbona et al., 2009*; *Pincus et al., 2010*;

*Rodriguez et al., 2012*; *Eletto et al., 2014*; *Morita et al., 2017*), since wIRE1α attenuation is significantly delayed but not completely prevented during prolonged ER stress.

An alternative possibility for the observed phenotypes of wIRE1α and sIRE1α is that these mutations in IRE1α affect IRE1α homo-oligomerization and/or ER stress-dependent activation of IRE1α independently of Sec61. Although future work is required to rule out this possibility, current results strongly indicate that the Sec61 translocon limits IRE1α oligomerization since two independent wIRE1α mutants exhibit similar effects and increased oligomerization. Furthermore, sIRE1α exhibits the opposite phenotype of wIRE1α, with reduced IRE1α oligomerization and slow activation/quick de-activation kinetics compared to wild-type IRE1α.

Upon first glance, the effects of disrupting the IRE1α-Sec61 interaction on XBP1u mRNA cleavage observed by our previous study (*Plumb et al., 2015*) and the current study may appear contradictory. However, the data are reconciled by considering the number of activated IRE1α molecules under all conditions. When an equal number of IRE1α and wIRE1α proteins are activated, wIRE1α exhibits less XBP1u mRNA cleavage because its RNase domains cannot access XBP1u mRNA as efficiently (*Plumb et al., 2015*), and thus produces less XBP1s protein (*Figure 5A,C and* five hour treatment). However, as we demonstrate in this study, disrupting the Sec61-IRE1α interaction results in a slight increase in the number of activated IRE1α molecules under normal conditions and a more dramatic increase during prolonged ER stress conditions (*Figures 4* and *5*). In this case, the difference in the ability of IRE1α and wIRE1α to access XBP1u mRNA becomes negligible, since so many more wIRE1α molecules are active compared to wild type IRE1α. Thus, the production of XBP1s protein is in fact greater in wIRE1α than wild type IRE1α expressing cells under such conditions.

Our data suggest that the intensity of ER stress determines whether IRE1α signaling is attenuated or remains active. We propose that the selective attenuation of IRE1α signaling may be beneficial for secretory cells such as pancreatic beta cells and plasma cells by providing a longer time window to resolve ER stress and avert inappropriate cleavage of ER-localized mRNAs, including mRNAs encoding secretory proteins such as insulin and immunoglobulin (*Lipson et al., 2006*; *Benhamron et al., 2014*). If ER stress is prolonged and irremediable, as shown by previous studies (*Lin et al., 2007*; *Rutkowski et al., 2006*; *Lu et al., 2014*), the PERK pathway remains active and induces CHOP-mediated cell death. However, severe ER stress may induce higher-order oligomers of IRE1α by overcoming the Sec61 translocon barrier, thus leading to a defect in the attenuation of IRE1α signaling. This continuous IRE1α activation might be beneficial for tumor growth (*Cubillos-Ruiz et al., 2015*) but may promote cell death in secretory cells such as pancreatic beta cells (*Ghosh et al., 2014*). In conclusion, the Sec61 translocon plays an essential role in controlling oligomerization and activity of IRE1α during ER stress. Thus, the IRE1α and the Sec61 translocon may be a prime target for small molecule manipulation to either enhance or suppress IRE1α signaling in diseases conditions.

# Materials and methods

### Antibodies and reagents

Antibodies were purchased: anti-FLAG (F3165, Sigma, St Louis, MO, CloneM2, RRID:AB_259529), anti-FLAG (L5) (637303, Bio-Legend, San Diego, CA, RRID:AB_1134265), anti-HA (MMS-101P, Covance clone 16B12, RRID:AB_2314672), anti-IRE1α (3294, Cell Signaling, Danvers, MA, RRID:AB_823545), anti-PERK (3192, Cell Signaling, RRID:AB_2095847), anti-IRE1α (20790, Santa Cruz, Dallas, Texas, RRID:AB_2098712), anti-Tubulin (ab7291, Abcam, Cambridge, UK, RRID:AB_2241126), anti-XBP1s (658802, BioLegend, RRID:AB_2562960), anti-BiP/GRP78 (610979, BD Biosciences, Franklin Lakes, NJ, RRID:AB_398292). Anti-HA, anti-Sec61α, and anti-Sec61β were gift from Dr. Ramanujan Hegde. Anti-mouse Goat HRP (11-035-003, Jackson Immunoreserach), anti-rabbit Goat HRP (111-035-003, Jackson Immunoreserach, RRID:AB_2313567), anti-Rb Cy3 (711-165-152, Jackson Immuno Research), anti-Mo Cy3 (715-165-150, Jackson Immuno Research, West Grove, PA, RRID:AB_2307443) and anti-Mo Cy2 (115-225-207, Jackson Immuno Research, RRID:AB_2338749).

Resins were purchased: anti-FLAG M2 affinity resin (A2220, Sigma-Aldrich, RRID:AB_10063035), anti-HA agarose (11815016001, Roche, Basel, Switzerland, RRID:AB_390914), anti-HA magnetic beads (88836, Fisher scientific, Waltham, MA), Strep-TactinXT beads (2-4010-010 IBA), SP Sepharose beads (17-0729-01, GE Healthcare, Chicago, IL).

Reagents were purchased: DMEM (10–013-CV, Corning, Corning, NY), FBS (16000044, Gibco, Gaithersburg, MD), Horse Serum (H0146, Sigma, St Louis, MO), Penicillin/Streptomycin (15140122, Gibco, ), Lipofectamine 2000 (11668019, Invitrogen, Carlsbad, CA), Doxycycline (631311, Clontech, Mountain View, CA) Hygromycin (10687010, Invitrogen), Blasticidin (ant-bl-1, InvivoGen), Thapsigargin (BML-PE180-0005, Enzo Life Sciences, Farmingdale, New York), Tunicamycin (T7765, Sigma), Protease inhibitor cocktail (11873580001, Roche), Biotin (B4639, Sigma), Digitonin (300410, EMD Millipore, Billerica, Massachusetts), Fluoromount G (0100–01, SouthernBiotech, Birmingham, AL), Phos-tag (300–93523, Wako, Japan), 3–12% BN-PAGE Novex Bis-Tris Gel (BN1003BOX, Invitrogen), SuperSignal West Pico or Femto Substrate (34080 or 34095, Thermo Scientific), $dCTP^{P32}$ (BLU013H001MC, PerkinElmer, Waltham, MA). All other common reagents were purchased as indicated in the method section.

## DNA constructs

pcDNA5/FRT/TO (Invitrogen, Carlsbad, CA) containing IRE1α-HA, wIRE1α (IRE1α-Δ434–443 HA) and IRE1α-K907A-HA were described previously (*Plumb et al., 2015*). IRE1α-T446A-S450A-T451A-HA mutant was created using previously described primers (*Sun et al., 2015*). sIRE1α (IRE1α-S439A-T446A-S450A-T451A-HA), IRE1α-V437A-D443A-HA, IRE1α-Δ434-443A-K907A-HA, in pcDNA5/FRT/TO were made by site-directed mutagenesis. Prl-His-2xstrep-IRE1α-FLAG constructs were generated by first inserting Prl-His-2xstrep into pcDNA5/FRT/TO using standard methods. Next, IRE1α-FLAG was amplified beginning from amino acid 29 and cloned into pcDNA5/FRT/TO Prl-His-2xstrep. Mouse spliced XBP1 plasmid (Addgene# 21833) is a kind gift from Dr. David Ron. All PCR reactions were performed with Phusion high fidelity DNA polymerase (NEB, Ipswich, MA), except for site directed mutagenesis, which used Pfu-Ultra polymerase (Agilent technologies, Santa Clara, CA). 3% DMSO was included in all PCR reactions to enhance amplification. The coding regions of all constructs were sequenced to preclude any sequence error. The Yale Keck DNA Sequencing Facility performed all sequencing services.

## Cell culture

HEK 293-Flp-In T-Rex cells were purchased from Invitrogen and cultured in high glucose DMEM (Corning, Corning, NY) containing 10% FBS (Gibco, Gaithersburg, MD ), 100 U/ml penicillin and 100 µg/ml streptomycin (Gibco) at 5% $CO_2$. IRE1α−/− HEK293-Flp-In T-Rex cells were previously described (*Plumb et al., 2015*). To establish stable cell lines, IRE1α−/− HEK293 cells were transfected with 1 µg of pOG44 vector (Invitrogen) and 0.1 µg of FRT vectors containing IRE1α or its mutants using Lipofectamine 2000 (Invitrogen). After transfection, cells were plated in 150 µg/ml hygromycin (Invitrogen) and 10 µg/ml blasticidin (InvivoGen, San Diego, CA). The medium was replaced every three days until colonies appeared. The colonies were picked and equal expression of the recombinant IRE1α or its mutants was evaluated by western blotting. The same protocol was applied in HEK 293-Flp-In T-Rex cells to generate Prl-His-2xstrep-IRE1α-FLAG stable cell lines of IRE1α, wIRE1α or sIRE1α. IRE1α −/−/FRT MEF cells (*Hollien et al., 2009*) are from Julie Hollien (University of Utah, USA) and they were complemented with either IRE1α, or wIRE1α as previously described (*Plumb et al., 2015*). INS-1 cells are from Richard Kibbey (Yale School of Medicine, USA) and were grown in RPMI (Sigma), 12.5% FBS (Gibco), 1 mM sodium pyruvate, 10 mM HEPES, 2 mM glutamine, and 50 µM and beta-mercaptoethanol. All the cell lines used in this study were not tested for mycoplasma, but many cell lines were used in immunofluorescence assays with Hoechst staining that should reveal presence of mycoplasma. Cells were assumed to be authenticated by their respective suppliers and were not further confirmed in this study. However, IRE1α knock out cell lines were verified by immunoblotting with IRE1α antibodies.

## ER stress treatment

Cells were counted and plated in 24 well ($1.5 \times 10^5$) plates and grown overnight to reach a confluence of 70% prior to treatment. In the case of overexpression study, doxycycline was added overnight. ER stress was induced by treating cells with tunicamycin (TM) or thapsigargin (Tg). All the concentrations and treatment time were as indicated in either result or figure sections. After the treatment, cells were directly harvested by adding 100 ul of 2X SDS sample buffer and boiled for 5 min with intermittent mixing and analyzed by western blotting. For XBP1u mRNA splicing assay, the

cells were harvested in Trizol (Ambion, Foster City, CA) and the splicing assay was performed as described previously (*Calfon et al., 2002*).

## BN-PAGE immunoblotting

Cells were lysed using 2% digitonin buffer (50 mM BisTris pH 7, 1x protease inhibitor cocktail [Roche], 100 mM NaCl and 10% Glycerol) for 45 min. Samples were than diluted to a final concentration of 1% digitonin and 50 mM NaCl. Samples were pelleted at 20,000$g$ for 20 min using refrigerated centrifuge. Supernatant was collected, mixed with BN-PAGE sample buffer (Invitrogen) and 5% G520 (Sigma). To run purified protein, samples were mixed in 1% digitonin buffer (50 mM BisTris pH 7, 1x protease inhibitor, 50 mM NaCl and 10% Glycerol) with BN-PAGE sample buffer and 5% G520.

Samples were run using 3–12% BN-PAGE Novex Bis-Tris (Invitrogen) gel at 150 V for 1 hr with dark blue buffer (50 mM Tricine pH 7, 50 mM BisTris pH 7% and 0.02%% G250) at room temperature and then exchange with light blue buffer (50 mM Tricine pH 7, 50 mM BisTris pH 7% and 0.002%% G250) for 4 hr in the cold room. To probe the Sec61 translocon, the gels were run for 1 hr with dark blue buffer at room temperature and 2 hr 45 min with light blue buffer in the cold room. After electrophoresis, gel was gently shaken in 1x Tris-Glycine-SDS transfer buffer for 20 min to remove residual blue dye. Transfer was performed using PVDF membrane (EMD Millipore) for 1 hr and 30 min at 85V. After transfer, the membrane was fixed with 4% acetic acid and followed with a standard western blotting procedure.

## Phostag assay

IRE1$\alpha$ phosphorylation was detected by previously described method (*Yang et al., 2010*). Briefly, 5% SDS PAGE gel was made containing 25 µM Phos-tag (Wako). SDS-PAGE was run at 100 V for 2 hr and 40 min. The gel was transferred to nitrocellulose (Bio-Rad, Hercules, CA) and followed with western blotting. The intensities of the Phos-tag bands were quantified with Image Quant TL software (GE HealthCare).

## Western blotting

Protein extracts were electrophoresed under reducing conditions on Tricine (Sigma) based SDS-PAGE gel and electro blotted onto nitrocellulose membrane (Bio-Rad). Blots were incubated with primary antibodies prepared in 1XPBS/Tween containing 5% BSA/0.02% NaN$_3$ for 1 hr and 30 min at room temperature. The secondary antibodies prepared in 5% Milk with 1XPBS/Tween were incubated for 1 hr at room temperature. Proteins were detected with SuperSignal West Pico or Femto Substrate (Thermo Scientific), exposed to Film BioExcel (Worldwide Life Sciences, Irvine, California) and developed.

## 2x Strep IRE1$\alpha$ and associating Sec61 complex protein purification

Stable cell lines expressing 2xStrep IRE1$\alpha$, wIRE1$\alpha$ and sIRE1$\alpha$ were induced with 200 ng/ml doxycycline and grown in 15 cm plate until 100% confluence. Cells were pelleted and proceed with microsome preparation as described (*Plumb et al., 2015*).. Briefly, cells were re-suspended in buffer (10 mM Hepes pH7.4, 250 mM Sucrose, 2 mM MgCl$_2$ and 1x protease inhibitor cocktail (Roche) and lysed by passing through 25-gauge for three times followed by 27-gauge for five times in cold room. Lysed samples were spun at low speed 2800$g$ for 30 min and supernatant was collected and spun at 75,000$g$ for 1 hr at 4℃ using MLA80 rotor. Microsome pellet was re-suspended in buffer containing (50 mM Hepes pH7.4, 250 mM Sucrose, 2 mM MgCl$_2$ and 0.5 mM DTT) and homogenized carefully using 2 ml dounce. Microsome concentrations were measured using absorbance A$_{280}$ and flash-freeze stored at −80℃ until further analysis.

Microsomes were lysed using 2% digitonin containing buffer (50 mM Tris pH8, 400 mM NaCl, 5 mM MgCl$_2$, 2 mM DTT, 1x protease inhibitor cocktail and 10% glycerol) for 1 hr at 4℃. Lysed microsomes were than diluted 1x with the same buffer omitting salt and digitonin and spun at 25 000$g$ for 30 min at 4℃ using MLA80 rotor. Supernatant was collected and proceed with protein purification. Briefly, supernatant was added to 10% vol pre-washed Strep-TactinXT beads (IBA) and rotated for 2 hr in cold room. Flow-through was removed and beads was transferred into 2 ml Bio-Rad column and washed with 10x beads volume using wash buffer (50 mM Tris pH8, 150 mM NaCl, 2 mM MgCl$_2$, 10% Glycerol and 0.2% digitonin). 2xStrep IRE1$\alpha$ was eluted from the beads using 50 mM

biotin (Sigma) buffer (50 mM Tris pH8, 150 mM NaCl, 2 mM MgCl$_2$, 10% Glycerol and 0.4% digitonin). Purified IRE1$\alpha$ and its associating Sec61-transclocon complex were further subjected to coomassie staining and quantified using BSA standards (Sigma).

To remove free IRE1$\alpha$, which is not bound to Sec61, the material was further purified by passing through SP Sepharose beads (GE Healthcare). Briefly beads were prepared in 2 ml Bio-Rad column and washed 5x using no salt buffer (20 mM Tris pH8, 2 mM MgAc and 0.4% digitonin). Purified protein was diluted 5x with no salt buffer and pass-through S-column. Beads were washed 5x column volume and eluted with 500 mM NaCl buffer (50 mM Tris pH8, 2 mM MgAc, 10% glycerol, and 0.4% digitonin). Purified IRE1$\alpha$-translocon complex was quantified along with BSA standards.

## In vitro XBP1 transcription and cleavage assay

1 µg of PCR purified XBP1u cDNA was transcribed using a master mix (1X RNA polymerase buffer (NEB), 0.4xNTP mix (Roche), mRNA cap (NEB), 0.01mCi $^{P32}$CTP (PerkinElmer), 8U RNasin (Promega, Madison, WI) and 20 U/ul SP6 enzyme (NEB). Transcription was performed at 40°C for 2 hr. XBP1u mRNA was extracted using Trizol reagent (Ambion) according to the manufactures method and dissolved in 100 µL pure water. XBP1u mRNA concentration was measured using absorbance A$_{280}$.

For the mRNA cleavage assay, purified IRE1$\alpha$ (5 nM) was mixed with cleavage buffer (50 mM Tris pH8, 50 mM NaCl, 5 mM MgCl$_2$, 0.4% digitonin, 1 mM ATP, 2 mM DTT and 2U RNAsin). The reaction was initiated by adding 2 ng of $^{P32}$CTP labeled XBP1u mRNA. Samples were incubated at 30°C. At each time point, sample was collected and the reaction was stopped by incubating at 70°C for 10 min in formamide (American Bioanalytical) sample loading buffer containing 5 mM EDTA and 1x bromophenol blue. Sample was loaded in a 6% Urea PAGE gel. Prior to actual samples running, gel was pre-run at 20W for 25 min. Actual sample running was performed at 9W for 35 min. Gel was fixed in 10% (methanol and acetic acid) for 20 min, dried at 55°C for 1 hr and 30 min. Gel was exposed to film and developed.

## Immunoprecipitation

To test the interaction between recombinant IRE1$\alpha$ and the endogenous Sec61 translocon, HEK 293 cells were transiently transfected with HA-tagged IRE1$\alpha$ constructs and expression induced with 100 ng/ml doxycycline. 24 hr after transfection, cells were harvested in 1xPBS and centrifuged for 2 min at 13,800$g$. Cell pellet was lysed in Buffer A (50 mM Tris pH 8, 150 mM NaCl and 1% digitonin) by rotating 30 min at 4°C. The supernatant was collected by centrifugation at 20,000$g$ for 15 min. For co-immunoprecipitation, supernatant was incubated with anti-HA-agarose (Roche) and anti-HA magnetic beads (Thermo Scientific). The beads were washed 3x with 1 ml of Buffer A containing 0.2% digitonin. The bound material was eluted from the beads by directly boiling in 50 µl of 2x SDS sample buffer and analyzed by immunoblotting.

## Immunostaining assay

Cells (0.12 $\times$ 10$^6$) were plated on 12 mm round glass coverslips (Fisher Scientific) coated with 0.1 mg/mL poly-lysine in 24-well plates. Expression of IRE1$\alpha$ constructs was induced with doxycycline (2 to 5 ng/ml) for 16 hr prior to treatment with ER stress inducers. For immunostaining, cells were fixed with 3.7% formaldeyhyde (J.T. Baker, Phillipsburg, New Jersey) for 10 min and permeabilized with 0.1% Triton X-100 (American Analytical, Akron, OH) for 5 min. The non-specific binding sites were blocked with Buffer A (1xPBS containing 10% Horse Serum and 0.1% Saponin) for 45 min. 100 µL of rabbit anti-HA, mouse anti-HA (Covance, Princeton, NJ), or anti-Sec61$\beta$ primary antibodies were added at 1:100 dilution in Buffer A and incubated for 1 hr, then washed 5X for 5 min. 100 µL of the secondary antibodies anti-rabbit Cy3, anti-mouse Cy3, and anti-mouse Cy2 (Jackson Immuno Research) were added at 1:100 dilution in Buffer A and incubated for 1 hr before washing five times with Buffer A. Coverslips were then incubated with 5 µg/ml Hoechst stain in 1xPBS for 15 min, washed with 1xPBS, and mounted using Fluoromount G (SouthernBiotech).

Cells were imaged on Leica scanning confocals (provided by the West Campus Imaging Core and the Nanobiology Institute at Yale University) consisting of an inverted microscope (Leica SP6/SP8), and an HC PL APO 63X (CS2 No: 11506350) oil objective lens (Leica, Wetzlar, Germany), and was controlled by the Leica Application Suite X. Sequential image scanning at 1x zoom, 100 Hz, 1024 $\times$

1024 pixels, and with line averaging set at four was used to collect images for cluster analysis. Sequential image scanning at 1.5x zoom, 100 Hz, and 2048x2048 pixels and line averaging of 6 was used for displayed images. To quantify number of cells with IRE1α puncta, the total number of cells per frame was first determined by manually counting Hoechst-stained nuclei. Only cells with a clearly present ER signal were included in this count. Subsequently, the number of cells with IRE1α puncta were counted, with puncta being defined as concentrated fluorescence signal typically approximately 0.4 um in diameter (4 hr tunicamycin treatment, 0.5 hr thapsigargin treatment) or approximately 1.5 μm in diameter (2 hr thapsigargin treatment). FIJI was used for cell counting. Data was graphed using GraphPad Prism and represented with standard error of the mean.

## Acknowledgements

We are grateful to Ramanujan Hegde for providing Sec61α, Sec61β, and HA antibodies. We thank the Mariappan lab for useful discussions. RP was supported by the CMB training grant from NIH (T32 GM007223). AS was supported by the Rudolph J Anderson Postdoctoral Fellowship. MM is supported by the Yale School of Medicine start-up package and NIH 1R01GM11738601.

## Additional information

### Funding

| Funder | Grant reference number | Author |
| --- | --- | --- |
| National Institutes of Health | NIH 1R01GM117386-01 | Rachel Plumb<br>Suhila Appathurai<br>Malaiyalam Mariappan |
| Yale School of Medicine | Start-up | Arunkumar Sundaram<br>Rachel Plumb<br>Suhila Appathurai<br>Malaiyalam Mariappan |
| Yale School of Medicine | Rudolph J Anderson Fellowship | Arunkumar Sundaram |
| National Institutes of Health | T32 GM007223 | Rachel Plumb |

The funders had no role in study design, data collection and interpretation, or the decision to submit the work for publication.

### Author contributions

AS, RP, Data curation, Formal analysis, Validation, Investigation, Methodology, Writing—review and editing; SA, Data curation, Formal analysis, Investigation, Methodology; MM, Conceptualization, Data curation, Formal analysis, Supervision, Funding acquisition, Validation, Methodology, Writing—original draft, Project administration, Writing—review and editing

### Author ORCIDs

Rachel Plumb, http://orcid.org/0000-0001-5329-5947
Malaiyalam Mariappan, http://orcid.org/0000-0002-2966-1182

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
