## [Decision Letter]

Thank you for submitting your article "The Sec61 translocon limits Ire1α signaling during the unfolded protein response" for consideration by *eLife*. Your article has been reviewed by two peer reviewers, and the evaluation has been overseen by a Reviewing Editor and Randy Schekman as the Senior Editor. The reviewers have opted to remain anonymous.

The reviewers have discussed the reviews with one another and the Reviewing Editor has drafted this decision to help you prepare a revised submission.

Summary:

This study from the Mariappan group is a follow-up to an earlier *eLife* contribution that described a physical interaction between Ire1α and the Sec61 translocon. The role of this interaction in the earlier study was proposed to be the efficient targeting of Xbp1 mRNA to Ire1α via the SRP-Sec61 targeting pathway. In this follow-up, the authors have exploited Ire1α mutants that either weaken (wIre1α) or strengthen (sIre1α) the Ire1α-translocon interaction to examine the consequences for ER stress-mediated Ire1α signaling. The central finding, shown in Figure 5, is that wIre1α is partially active even without ER stress, and fails to be efficiently turned off during chronic stress. By contrast, sIre1α is slightly more efficiently turned off relative to wild type Ire1α. This is a carefully done study and a well-written paper including interesting and important results. However, some points need clarifying before publishing.

Essential revisions:

1) The three reviewers all indicated that the authors should be much more cautious in their interpretation of the mobilities of membrane protein complexes on the Blue-native gels. Mobility of membrane proteins on these gels is impacted by shape and bound detergent. The molecular weight markers used for the gels are not membrane proteins, so they don't bind the detergent. Previous work from the Skach lab has characterized the mobility of a Sec61-heterotrimer on BN-PAGE gels. For example, the 66 kD band of Sec61 (panel E) is almost certainly not a Sec61 heterotrimer, but is instead just Sec61α. (The Sec61 heterotrimer should migrate at ~240 kD, see Conti et al. Mol. Cell 58:269). For this reason, the number of Sec61 and Ire1 proteins in the A and B bands (Figure 1), is almost certainly smaller than the authors suggest in the Discussion. The Blue Native gel data can remain in a revised version of the manuscript, but these results should only be used to show that at a gross level, oligomerization/interaction status does not change appreciably upon ER stress.

2) The central conclusion is predicated on the assumption that the Ire1α mutants only affect its interaction with Sec61, and nothing else. The observation of opposite effects for wIre1α and sIre1α helps make this argument, as does the fact that multiple wIre1α alleles seem to have similar effects. Nevertheless, it seems important to discuss the alternative possibility that this region of Ire1α is critical for homo-oligomerization and/or stress-mediated activation independently of Sec61 interaction, and this might explain the findings. The best way to rule this out experimentally would be to make mutants in Sec61 that disrupt its ability to bind to Ire1α, and show that this mimics the wIre1α phenotype. However, this is beyond the scope of this study. The Sec61a knockdowns partially address this, but this approach is not explored much, which is understandable because the cells probably have many other things wrong with them. This issue should be addressed in the Discussion.

3) The functionality of the Ire1α mutants in the Xbp1 splicing assay (Figure 2—figure supplement 1) seems different from wild type. Is this reproducible and/or relevant? This relates to point 1, where one must be cautious about unintended effects of the mutants.

4) The authors demonstrated that Ire1α formed cluster under severe stress condition (Figure 6). Under this condition, there is a possibility that Ire1α would dissociate from Sec61 and free form of Ire1α could form cluster like wIre1α. The authors should clarify or discuss whether Sec61-associated Ire1α forms cluster or not. Immunofluorescence experiments using anti-Sec61 should be done to address this question in stress treated cells.

5) The authors describe that ER stress is required for the activation of preformed Ire1α complex. However, overexpression of Ire1α spontaneously activate Ire1α molecules. How do the authors explain it?

6) In their first paper (and independently discovered by Kohno group), Sec61 interaction facilitates Xbp1 splicing; now, they show that disrupting this interaction with their wIre1 allele facilitates Xbp1 splicing and in fact, results in mild constitutive Ire1 activation. Some discussion of this apparent contradiction is warranted.

7) The authors need to cite additional papers and comment on the role of BiP binding in activation of Ire1. In the regulation of Ire1 activation, two molecules have been well studied. One is BiP and the other is misfolded or unfolded proteins. In this manuscript, the authors found the Sec61 is also an important molecule. There is little discussion about the relationship between these molecules and Sec61. If the authors claim the importance of Sec61, they had better discuss the relationship among them, especially BiP, if possible.

In the last paragraph of the Introduction, the authors described BiP is an important molecule and they cited recent papers, however, they skipped the first important reports (mammalian cells: Bertolotti et al., Nat Cell Biol, 2000; yeast cells, Okamura et al., BBRC, 2000).

In the fifth paragraph of the Discussion, the work by Peter Walter's group (Gardner et al., Science, 2011) is cited, but the first work about the direct interaction between unfolded proteins and Ire1 reported by Kimata (Kimata et al., JCB, 2007) is not cited.

---

## [Author Response]

*Essential revisions:*

*1) The three reviewers all indicated that the authors should be much more cautious in their interpretation of the mobilities of membrane protein complexes on the Blue-native gels. Mobility of membrane proteins on these gels is impacted by shape and bound detergent. The molecular weight markers used for the gels are not membrane proteins, so they don't bind the detergent. Previous work from the Skach lab has characterized the mobility of a Sec61-heterotrimer on BN-PAGE gels. For example, the 66 kD band of Sec61 (panel E) is almost certainly not a Sec61 heterotrimer, but is instead just Sec61α. (The Sec61 heterotrimer should migrate at ~240 kD, see Conti et al. Mol. Cell 58:269). For this reason, the number of Sec61 and Ire1 proteins in the A and B bands (Figure 1), is almost certainly smaller than the authors suggest in the Discussion. The Blue Native gel data can remain in a revised version of the manuscript, but these results should only be used to show that at a gross level, oligomerization/interaction status does not change appreciably upon ER stress.*

We appreciate the reviewers for pointing out the issues with the blue-native PAGE gels. As they suggested, we have removed the section in the Discussion describing the possible number of Sec61 and Ire1α molecules in A and B bands. We have now used the data to show only that Ire1α oligomerization is not appreciably changed upon ER stress and that oligomerization differs when the Ire1α-Sec61 interaction is disrupted. We apologize that in our initial manuscript we mistakenly labeled the Sec61 translocon as a 66 kDa band due to the difficulty in detecting the lower molecular weight marker proteins after transferring onto the PVDF membrane. We have now performed a careful experiment to validate the size of the Sec61 translocon on the BN-PAGE gels by comparing with a protein standard lane from the same BN-PAGE gel that was excised and stained with Coomassie blue (Figure 7). We now see both Sec61a and Sec61b migrating as a ~146 kDa band. This suggests that the Sec61 translocon complex is intact and is also consistent with Conti et al., where they see the majority of the Sec61 translocon (α, β, γ) migrating as a ~130kDa band (Conti et al., Figure 1). The reviewers pointed to a ~240 kD band that corresponds to the Sec61 heterotrimer in Conti et al., but we were not able to find this complex in their data or Discussion. Interestingly, we detect a ~350 kDa band with the Sec61b antibodies and a faint band at 350kDa for Sec61a, consistent with Conti et al., where they identified this band as a Sec61 and TRAP complex. Collectively, these results suggest that the migration of the Sec61 translocon we observe is consistent with the previous findings from the Skach lab.

Author response image 1.BN-PAGE immunoblotting analysis of the Sec61 translocon complex.IRE1α -/- HEK293 cells complemented with wild-type IRE1α-HA or wIREα-HA were treated with 2.5 μg/ml Tg for the indicated hours (hr), lysed with digitonin, and analyzed by BN-PAGE immunoblotting as well as a protein marker lane from the same BN-PAGE gel was excised.**DOI:**
http://dx.doi.org/10.7554/eLife.27187.027

*2) The central conclusion is predicated on the assumption that the Ire1α mutants only affect its interaction with Sec61, and nothing else. The observation of opposite effects for wIre1α and sIre1α helps make this argument, as does the fact that multiple wIre1α alleles seem to have similar effects. Nevertheless, it seems important to discuss the alternative possibility that this region of Ire1α is critical for homo-oligomerization and/or stress-mediated activation independently of Sec61 interaction, and this might explain the findings. The best way to rule this out experimentally would be to make mutants in Sec61 that disrupt its ability to bind to Ire1α, and show that this mimics the wIre1α phenotype. However, this is beyond the scope of this study. The Sec61a knockdowns partially address this, but this approach is not explored much, which is understandable because the cells probably have many other things wrong with them. This issue should be addressed in the Discussion.*

We have now discussed the alternative possibility that the Sec61 translocon interaction region in Ire1α might also be crucial for Ire1α homo-oligomerization and/or ER stress mediated activation. As reviewers suggested, we had previously attempted to find the Ire1α interacting region in Sec61a. Since the Sec61 translocon interaction region is present in the luminal domain of Ire1α, we predicted that a luminal of loop of Sec61a might be important for the interaction with Ire1α. However, we encountered a problem as some of the deletion constructs of Sec61 exhibit reduced expression and poor assembly with the endogenous Sec61b subunit, thus confounding the analysis of our data (Figure 8).

Author response image 2.Domain mapping of the IRE1α interaction region in Sec61α.The lysates prepared from HEK293 cells transiently expressing the indicated 3x-HA tagged Sec61a variants were immunoprecipitated with anti-HA antibodies and analyzed by immunoblotting. Δ denotes deletion of indicated amino acid residues from canine Sec61a.**DOI:**
http://dx.doi.org/10.7554/eLife.27187.028

*3) The functionality of the Ire1α mutants in the Xbp1 splicing assay (Figure 2—figure supplement 1) seems different from wild type. Is this reproducible and/or relevant? This relates to point 1, where one must be cautious about unintended effects of the mutants.*

We reproducibly observed that both wIre1α and sIre1α proteins showed slightly less Xbp1 mRNA cleavage compared to the wild type. This data is consistent with the reduction in Xbp1s protein production for both sIre1α and wIre1α. However, since Xbp1 mRNA targeting to Ire1α should not be a factor in the in vitro cleavage assay, the reduction in wIre1α cleavage is somewhat surprising. It is likely that the lack of other components in this assay, such as misfolded protein activating ligands and BiP, prevents an accurate comparison between the Xbp1 mRNA cleavage observed in vitro and the Ire1α activity demonstrated in cells. We agree with the reviewers concerns about unintended effects, and now include the possibility of alternative effects of Ire1α mutations in the Discussion.

*4) The authors demonstrated that Ire1α formed cluster under severe stress condition (Figure 6). Under this condition, there is a possibility that Ire1α would dissociate from Sec61 and free form of Ire1α could form cluster like wIre1α. The authors should clarify or discuss whether Sec61-associated Ire1α forms cluster or not. Immunofluorescence experiments using anti-Sec61 should be done to address this question in stress treated cells.*

As reviewers suggested, we have performed the co-localization of Ire1α with the Sec61 translocon using Sec61b antibodies in Ire1α, wIre1α or sIre1α expressing cells. Our results suggest that wild type Ire1α and sIre1α clusters formed under severe stress conditions co-localize with the endogenous Sec61beta (Figure 6—figure supplement 2). In contrast, wIre1α clusters were devoid of the Sec61beta staining. This result is consistent with co-immunoprecipitation experiments, where Ire1α always associated with the Sec61 translocon even during severe stress (Figure 6—figure supplement 1).

*5) The authors describe that ER stress is required for the activation of preformed Ire1α complex. However, overexpression of Ire1α spontaneously activate Ire1α molecules. How do the authors explain it?*

Ire1α overexpression can indeed activate more than 50% of Ire1α molecules (Figure 4), but its complete activation still requires ER stress. Moreover, Ire1α over expression alone is insufficient to induce clusters in cells, which strictly require ER stress treatment. Although the precise mechanism for the partial activation of overexpressed Ire1α is unclear, it may be that Ire1α overexpression results in the formation of many pre-formed complexes that may transiently collide and trigger trans-autophosphorylation independent of ER stress. Alternatively, it could be that overexpression of Ire1α may disturb ER homeostasis and activate Ire1α.

*6) In their first paper (and independently discovered by Kohno group), Sec61 interaction facilitates Xbp1 splicing; now, they show that disrupting this interaction with their wIre1 allele facilitates Xbp1 splicing and in fact, results in mild constitutive Ire1 activation. Some discussion of this apparent contradiction is warranted.*

We have described two independent functions of the Ire1α-Sec61 interaction in Ire1α signaling. In our first publication, we show that the maximal amount of Xbp1 mRNA cleavage during ER stress is less for wIre1α than wild-type Ire1α. In addition, Kenji Kohno’s group and we demonstrate that Xbp1 mRNA is targeted to the Sec61 translocon. Therefore, we conclude that the Ire1α-Sec61 interaction is necessary to bring together Ire1 and its primary mRNA substrate, Xbp1, for efficient cleavage during ER stress conditions. Consistent with this conclusion, in our current work we find that the maximal production of Xbp1s protein in wIre1α is reproducibly less than wild-type Ire1α (Figure 5 and Figure 6, see 5 hr treatment). It is important to point out that losing the interaction with Sec61 does not wholly abolish Xbp1 mRNA cleavage by Ire1α (Plumb and Zhang et al., 2015 and this study). In addition, we have noticed that the extent of the effect on Xbp1 mRNA cleavage is highly dependent on Ire1α and wIre1α expression levels; the closer to endogenous levels of Ire1α, the more dramatic the difference. Indeed, even a small increase in wIre1α expression can overcome this shortcoming.

Here, we describe an additional role for the Ire1α-Sec61 interaction, namely in limiting the extent of Ire1α oligomerization and the subsequent kinetics of its activation and de-activation, as shown by phosphorylation, during ER stress. Therefore, wIre1α activity is higher than wild type Ire1α under two different conditions. First, a small population of wIre1α is constitutively activated even under normal conditions. Second, we observe that wIre1α remains activated longer than Ire1α during prolonged ER stress (Figure 5). Under these conditions, the difference in the ability of Ire1α and wIre1α to access Xbp1 mRNA becomes negligible, since many more wIre1 molecules are active compared to wild type Ire1α. Thus, the production of Xbp1s protein is in fact greater in wIRE1α than wild type Ire1α expressing cells under such conditions. We have now included a section discussing this issue in the manuscript.

*7) The authors need to cite additional papers and comment on the role of BiP binding in activation of Ire1. In the regulation of Ire1 activation, two molecules have been well studied. One is BiP and the other is misfolded or unfolded proteins. In this manuscript, the authors found the Sec61 is also an important molecule. There is little discussion about the relationship between these molecules and Sec61. If the authors claim the importance of Sec61, they had better discuss the relationship among them, especially BiP, if possible.*

*In the last paragraph of the Introduction, the authors described BiP is an important molecule and they cited recent papers, however, they skipped the first important reports (mammalian cells: Bertolotti et al., Nat Cell Biol, 2000; yeast cells, Okamura et al., BBRC, 2000).*

*In the fifth paragraph of the Discussion, the work by Peter Walter's group (Gardner et al., Science, 2011) is cited, but the first work about the direct interaction between unfolded proteins and Ire1 reported by Kimata (Kimata et al., JCB, 2007) is not cited.*

We apologize for missing important papers in our reference list. We have included a section in the Discussion about the role of BiP in regulating Ire1α activity and cited appropriate references.